# Anisotropy in mechanical unfolding of protein upon partner-assisted pulling and handle-assisted pulling

Nisha Arora[1], Jagadish Prasad Hazra ![ORCID] [1]✉ & Sabyasachi Rakshit ![ORCID] [1,2]✉

Proteins as force-sensors respond to mechanical cues and regulate signaling in physiology. Proteins commonly connect the source and response points of mechanical cues in two conformations, independent proteins in end-to-end geometry and protein complexes in handshake geometry. The force-responsive property of independent proteins in end-to-end geometry is studied extensively using single-molecule force spectroscopy (SMFS). The physiological significance of the complex conformations in force-sensing is often disregarded as mere surge protectors. However, with the potential of force-steering, protein complexes possess a distinct mechano-responsive property over individual force-sensors. To decipher, we choose a force-sensing protein, cadherin-23, from tip-link complex and perform SMFS using end-to-end geometry and handshake complex geometry. We measure higher force-resilience of cadherin-23 with preferential shorter extensions in handshake mode of pulling over the direct mode. The handshake geometry drives the force-response of cadherin-23 through different potential-energy landscapes than direct pulling. Analysis of the dynamic network structure of cadherin-23 under tension indicates narrow force-distributions among residues in cadherin-23 in direct pulling, resulting in low force-dissipation paths and low resilience to force. Overall, the distinct and superior mechanical responses of cadherin-23 in handshake geometry than single protein geometry highlight a probable evolutionary drive of protein-protein complexes as force-conveyors over independent ones.

[1] Department of Chemical Sciences, Indian Institute of Science Education and Research Mohali, Mohali, Punjab, India. [2] Centre for Protein Science Design and Engineering, Indian Institute of Science Education and Research Mohali, Mohali, Punjab, India. ✉email: jagadish.hazra@gmail.com; srakshit@iisermohali.ac.in

Mechanical tension, as one of the critical cues in physiology, regulates several biological processes, including gene-expression[1–4], blood-coagulation[5], cell adhesion[6,7], muscle function[8,9], hearing[10–12], bacterial anchorage[13–15], and more. Biomacromolecules, especially proteins with unique viscoelastic properties, primarily serve as force-sensors or conveyers and orchestrate such mechanoresponsive processes. Interestingly, two configurations among the protein-based force-sensors/conveyers are commonly observed in physiology. In one, a single protein links the source and the response points of the mechanical cue in an end-to-end configuration and transmits further. Connectin protein that regulates contraction of striated muscle tissues[16,17], elastin in the extracellular (EC) matrix that imparts elasticity and resilience to tissues[18,19] fall in this category. The second configuration is more abundant in nature. The mechanical stimuli in this configuration are transmitted through protein-complexes where the protomers interact over an overlapping binding interface in a handshake configuration. Von-Willebrand factor interacts with the cell-surface glycoproteins of platelets in this configuration and facilitates blood coagulation under the mechanical cue from hydrodynamic shearing[20,21]. Nonclassical cadherins form heterophilic tip-link complexes in handshake configuration and transduce mechanical inputs in hearing and balance[22]. Other cadherins, too, form homophilic handshake complexes at the cell-cell junction and regulate morphogenesis[23]. Actomyosin complexes regulate mechanoresponsive cell-motility[24–26]. Interestingly, the function and evolution of protein-protein interactions in signal transduction are well versed, the evolutionary importance of such protein complexes in force-transduction is still elusive.

Conventionally, force spectroscopy (preferably at the single-molecule level) is utilized to decipher the thermodynamics, kinetics[27,28], and molecular mechanisms[29,30] of force-transduction through force-sensor proteins under in-vitro mechanical stimuli. In single-molecule force spectroscopy (SMFS), the protein of interest (POI) is either attached with marker polyproteins or DNA and pulled from one end with a mechanical spring. This can be described as 'handle (or hook)-assisted pulling' (HAP). In HAP, the mechanical spring-handle connects to the POI either specifically via thiol-bonds[31–33], non-covalent ligand-receptor interactions using biotin-streptavidin complexes[34], cohesin-dockerin complexes[35], Ni-NTA-His complexes[36], or non-specifically. Finally, the quantitative dependency of force-resilience on the intrinsic factors like the secondary-structure compositions, conformational entropy, short-range and long-range interactions, hydrogen-bond (H-bond) network, hydrophobic core, domain arrangements[37], etc. are deciphered from the unfolding/refolding force-extension relations. It is interesting to note that the force-response of proteins is also sensitive toward the directions of pulling. Anisotropy in the mechanical response of proteins like GFP[38,39], ankyrin[40], GB1[39,41], srcSH3[42], etc., are observed with the change in pulling direction or tethering geometry.

While SMFS undoubtedly enriched the mechanobiology of proteins, the HAP conveniently imitates the working model for type one configuration of force-sensors/conveyers. Therefore, the fundamental question, whether the HAP based force-spectroscopy is suitable to decipher the force-responsive nature of proteins in the second type of force-sensors, remains still elusive. Instead, the SMFS for the second type of force-sensing configuration is exclusively used to understand the force-resilience of the complexes but not for the constituent proteins. We define the second configuration of pulling in force spectroscopy as 'partner-assisted pulling' or PAP. Intuitively, PAP and HAP are geometrically different and expected to follow different force-resilience mechanisms. With an overarching objective of elucidating the evolutionary thirst for developing two configurationally different

force-sensors, here, we plan to experimentally decipher the difference in the force-responsive properties of HAP and PAP and highlight the underlying molecular mechanisms.

To elucidate the difference between HAP and PAP, we plan to use a natural force-sensor from the second configuration but with a strong binding affinity to partners. Tip-links that serve as gating-spring in hearing[43,44] are known to form strong adhesive interactions between constituent proteins, cadherin-23 (Cdh23) and protocadherin-15 (Pcdh15)[45]. These two proteins as tip-links receive tensile forces of varying magnitudes ranging from 10 to 100 pN from sound-stimuli and convey the force to ion channels during mechanotransduction in hearing. Interestingly, the lifetime of the tip-links complex is measured at varying tensile forces using PAP-based SMFS[46]; however, the force-responsive behaviour of the constituent cadherins has conveniently been measured using HAP configuration[47,48]. Whether the complex configuration of tip-link has any implication on the force-response of constituent cadherins is thus unclear. Here, we plan to utilize the strong binding affinity of the Cdh23-Pcdh15 to understand the mechanoresponsive behaviour of Cdh23 in PAP mode and decipher how PAP is different from HAP. To note, SMFS using AFM has obtained an off-rate of $4.5 \times 10^{-3} \pm 4.9 \times 10^{-5}\,\text{s}^{-1}$ and distance to the transition state of $0.18 \pm 0.03$ nm of the tip-link complex when measured using the two outermost domains of tip-link cadherins[49]. We notice higher mechanical resilience of Cdh23 during pulling in PAP mode compared to HAP. The aggravated mechanical response of Cdh23 in PAP may also have an impact on the evolution of protein-protein complexes as force-conveyers over a single spring protein.

## Results

**Design of HAP and PAP modes**. Being a nonclassical cadherin family of proteins, Cdh23 has 27 EC domains and Pcdh15 has 11 EC domains, apart from the transmembrane and cytosolic components. Two outermost EC domains at the N-terminals, EC1-2, of both cadherins overlap in a handshake geometry and form the tip-link complex[45]. For SMFS using HAP and PAP modes with an Atomic Force Microscopy (AFM), we use the entire EC region of Cdh23 (Cdh23 EC1-27). The C-terminus of Cdh23 is recombinantly modified with sortag (-LPETGSS) for covalent anchorage to glass-coverslips using sortase A[50] and expressed in mammalian Expi-CHO cell lines (Supplementary Fig. 1). For HAP, we have recombinantly modified the N-terminus of Cdh23 with Avitag (See Methods) for enzymatic conjugation of single biotin using BirA (Fig. 1a). For PAP, the N-terminus of Cdh23 is left unaltered; however, the AFM cantilever is covalently modified with Pcdh15 EC1-2 as a partner for pulling Cdh23 by utilizing tip-link complexation (Fig. 1b) (See Methods).

**HAP and PAP feature distinctly different unfolding patterns of Cdh23**. Single-molecule pulling of polyproteins comprising repeats or identical domains in tandem feature sawtooth pattern of protein-unfolding where the unfolding peaks are evenly separated along the extension. Though Cdh23 possesses multiple EC domains in tandem, such well-behaved unfolding force-extension patterns are not observed in both HAP and PAP. This is expected for Cdh23 as the EC domains, though feature similar structural architectures with seven β-strands and connecting loops, possess diversity in constituent amino acid residues with sequence similarities <30%[51]. The variety is reported even in the interdomain linkers that are usually 11–12 residues long but vary in the amino acid sequences and their affinities toward $Ca^{2+}$-ions. It is important to note that the interdomain linkers in cadherins bind with $Ca^{2+}$-ions (three for canonical linkers)[52], and reduce the conformational entropy of the proteins. Cdh23

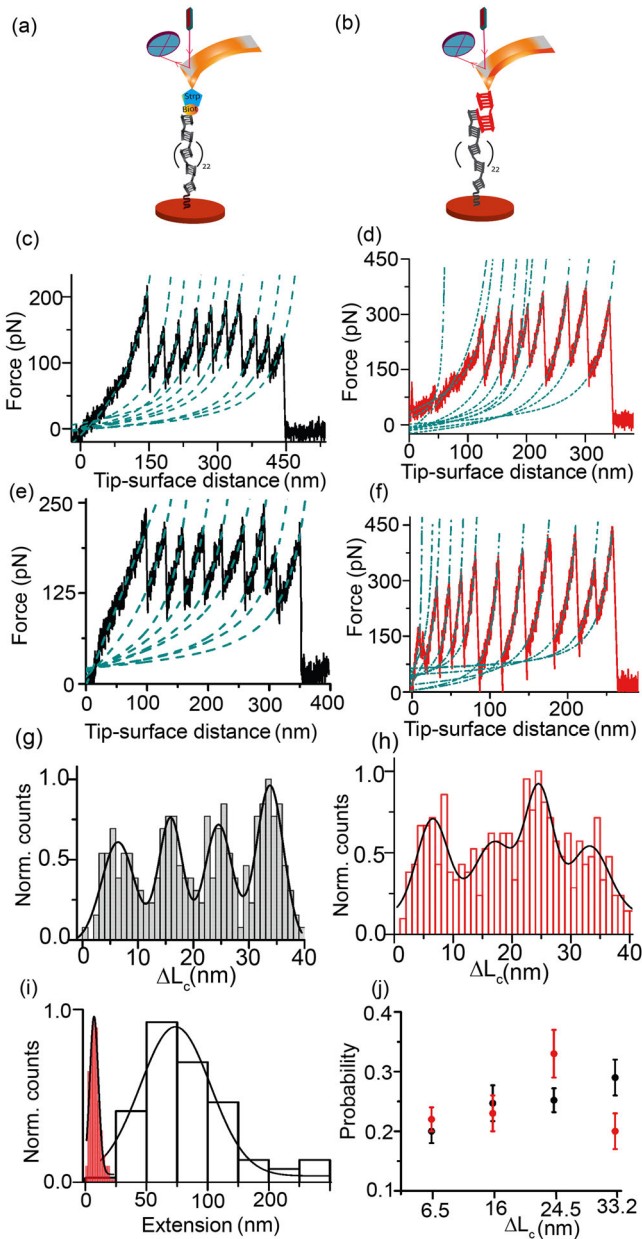

**Fig. 1 Distinct unfolding propensity of Cdh23 in HAP and PAP configurations. a** Schematic representation of HAP of Cdh23 using AFM where the C-terminus of Cdh23 is covalently attached to glass coverslip using sortagging. The N-terminal of Cdh23 is recombinantly modified with biotin and pulled by streptavidin-coated cantilevers. **b** Scheme of PAP configuration depicting the partner-assisted pulling of Cdh23 EC1-27 with Pcdh15 EC1-2 (red). **c** and **e** Representative force-extension curves of Cdh23 in HAP (black) at 2000 nm s⁻¹ and 3000 nm s⁻¹ pulling speeds, respectively. Both curves depict an initial long stretch followed by sawtooth patterns. **d** and **f** Representative force-extension features of Cdh23 in PAP (red) at 2000 nm s⁻¹ and 3000 nm s⁻¹ pulling speeds respectively. Both curves exhibit multiple unfolding peaks at the initial stretches, followed by sawtooth patterns of unfolding. The dotted green lines in (**c**-**f**) are the WLC model fit. **g** and **h** Distributions of $\Delta L_c$ of Cdh23-stretching in HAP (black line) ($n = 838$ unfolding events) and PAP (red) ($n = 946$ unfolding events), respectively. The data are combined for all loading rates. Solid lines represent the Gaussian fits exhibiting four unfolding peaks. **i** Distributions of initial long and short stretches of Cdh23 obtained in HAP (black) and PAP (red), respectively. The data are plotted for all the force curves obtained at different pulling velocities. Gaussian fittings of the histograms (solid black line) exhibit a mean length gain of $75.0 \pm 2.2$ nm in HAP and $6.5 \pm 0.3$ nm in PAP. Errors mentioned are the standard error of fitting. **j** Probability distributions of four different stretches of Cdh23 in HAP (black) and PAP (red).

silico pulling of a single domain of Cdh23 (Cdh23 EC1)[54]. Further, while analyzing individual force-extension curves, we notice differences in the unfolding patterns in the spectra for both HAP and PAP (Fig. 1c–f). Incidentally, the static distribution of $\Delta L_C$ fails to highlight differences; however, the probability distributions of each extension distinctly point out the differences in unfolding preference for HAP and PAP (Fig. 1j). Across all extensions, we observe a higher probability for a shorter extension of 6.5 nm for PAP. On the contrary, the probabilities of longer stretches of 24.5 nm and 33.2 nm are dominant in HAP. Further, HAP features an initial low-force (most-probable force, $F_{mp} = 52.4 \pm 3.2$ pN) extension of $75.0 \pm 2.2$ nm followed by a sawtooth pattern of unfolding (Fig. 1c, e, i, & Supplementary Fig. 4). The initial low-force stretching may correspond to the entropic extension of the protein. Contrary to HAP, multiple unfoldings with short extensions with a peak-maximum of $6.5 \pm 0.3$ nm is preferentially observed for PAP configuration at the initial low-force regime ($F_{mp} = 44.6 \pm 2.4$ pN) (Fig. 1i & Supplementary Fig. 4).

Generally, one residue contributes 0.38 nm to the extension of a protein. Accordingly, ~6.5 nm, ~16 nm, ~24.5 nm, and ~33 nm of extensions originate from the unfolding of 17, 42, 64, and 89 residues. On average, linkers and domains are comprised of 12 and 96 residues, respectively. Therefore, ~6.5 nm and ~33 nm extensions can intuitively be attributed to the unfolding of linker and domain, respectively. Likewise, the origins of ~16 nm and ~24.5 nm extensions cannot be assigned explicitly due to a lack of in-depth knowledge on the domain-wise unfolding. These extensions may arise from the partial unfolding of domains (intermediate states) or partial unfolding of both domain and linker.

To derive the quantitative kinetic models for all four $\Delta L_C$ extensions, we segregate the unfolding events based on $\Delta L_C$ and plot the comparative unfolding force-distributions with LR for both HAP and PAP (Fig. 2a–d). Raw data for force-distributions are provided in Supplementary Data 1. Here, too, the higher unfolding forces are observed for PAP for all $\Delta L_C$ (except for ~6.5 nm) at all LR. From the Bell–Evans model fit (Eq. 2, Methods)[55,56] to the $F_{mp}$ vs. *loading rate* plots (Fig. 2e–h, Supplementary Fig. 5), we determine the intrinsic unfolding transition rates ($k_u^0$) and the widths ($x_\beta$) of their potential barriers of the extensions, and further

possesses both canonical and noncanonical linkers. Noncanonical linkers lack the Ca²⁺-affinity and bind to two or less Ca²⁺-ions in physiological conditions, thus increasing the conformational entropy of proteins.

We observe multiple unfolding peaks from the single-molecule pulling of Cdh23 in both HAP and PAP (Fig. 1c–f and Supplementary Fig. 2). Interestingly, distinctly higher force resistance by Cdh23 is noticed in PAP configuration than HAP from the overall unfolding force-distributions at varying loading rates (LR) (Supplementary Fig. 3). To highlight the differential effects of pulling stereography on Cdh23, we measure the change in contour length ($\Delta L_C$) from the Worm-like chain (WLC) model fit to the force-extension curves and generate histograms of $\Delta L_C$ including all unfolding events for HAP and PAP[53] (Eq. 1, Methods). Four distributions of $\Delta L_C$ is observed for both HAP and PAP, with peak-maxima at $6.4 \pm 0.4$, $15.9 \pm 0.3$, $24.5 \pm 0.3$, $33.7 \pm 0.2$ nm and $6.5 \pm 0.5$, $16.9 \pm 1.4$, $24.6 \pm 0.6$, $33.2 \pm 0.9$ nm, respectively (Fig. 1g, h). Four distributions of $\Delta L_C$ indicate the multistep unfolding of Cdh23 in both HAP and PAP. It may be noted that the multistep unfolding is already reported from an in

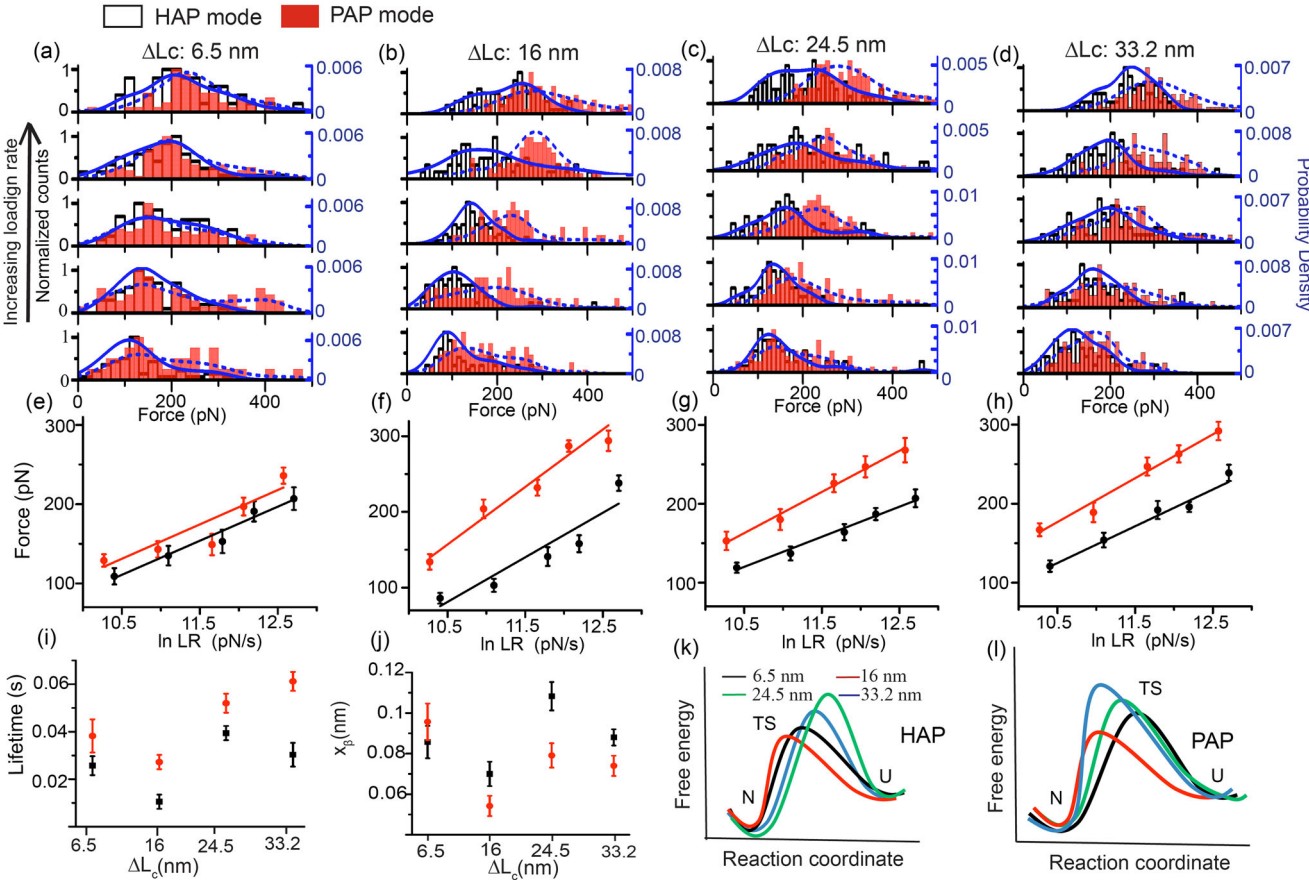

**Fig. 2 Cdh23 exhibits higher mechanical tolerance in PAP configuration compared to HAP.** The distributions of unfolding forces of Cdh23 at different pulling velocities (500, 1000, 2000, 3000, 5000 nm s$^{-1}$) have been plotted for four different extensions of (**a**) ~6.5 nm, **b** ~16 nm, **c** ~24.5 nm, and (**d**) ~33 nm for both HAP (black) and PAP (red). The solid and the dashed blue lines are the kernel density estimates (KDE) for HAP and PAP, respectively. The most probable unfolding forces are chosen from the maximum counts in the KDE. **e–h** The monotonous increase in the most probable unfolding forces with loading rates (loading-rate = pulling velocity × spring constant of the cantilever) is plotted for four different $\Delta L_{C}$s. Errors reported are the standard error. The solid lines are the Bell–Evans model fit to the force-loading rate data for (**e**) ~6.5 nm, **f** ~16 nm, **g** ~24.5 nm, and (**h**) ~33 nm. **i** and **j** Distributions of the intrinsic lifetime ($\tau_{off}$) and $x_{\beta}$ of all four extensions are shown respectively for HAP (black) and PAP (red) (mean ± SD from $n = 3$ independent experiments). **k** and **l** Schematics of potential energy barriers for extensions of Cdh23 of ~6.5 nm (black), ~16 nm (red), ~24.5 nm (green), and ~33 nm (blue) are drawn as per the kinetic parameters for HAP and PAP, respectively.

activation barrier height ($\triangle G^{*}$) of the escape-energy diagrams (Tables 1 and 2 and Supplementary Tables 1 and 2). We obtain higher $x_{\beta}$ and faster $k_{u}^{0}$ in HAP than PAP for $\Delta L_{C}$ of ~16 nm, ~24.5 nm and ~33 nm. While the faster-unfolding transition rates directly indicate easier escape in HAP (Fig. 2i), higher $x_{\beta}$ values (Fig. 2j) make the potential energy barrier of protein folding more susceptible to force in HAP (Fig. 2k, l, Supplementary Fig. 6). Overall, we measure four unfolding states of Cdh23 in both HAP and PAP; however, their relative populations and potential energy diagrams differ between HAP and PAP. We estimate the critical force ($F_{C}$) for unfolding from the kinetic parameters of protein folding, i.e., the force-equivalent to diminish the potential barrier of extensions. We notice the lowest $F_{C}$ for ~6.5 nm extension in PAP and ~24.5 nm extension in HAP, indicating two different most susceptible states in two pulling geometries. Incidentally, the distributions of $\Delta L_{C}$ do not directly reflect the same, and the discrepancy may arise from the assumption that the pulling direction (in both HAP and PAP) is always aligned with the distortion or the $x_{\beta}$. However, in likelihood, the pulling directions alter with extensions.

**Dynamic network structures of Cdh23 under tension differ for HAP and PAP.** PAP steers the unfolding of longer extensions

through a steeper and narrower energy barrier, thus makes proteins more resilient to large perturbations in response to force than the HAP. However, the molecular mechanisms that provide more mechanical stability in proteins in PAP configuration than HAP are still elusive. To decipher, we plan for all-atom steered molecular dynamics (SMD) simulations[57,58] mimicking experimental PAP and HAP methods and perform dynamic network analysis to identify the structural contributions in force-dissipation for both PAP and HAP[59]. We further construct the force-propagation pathways in the proteins in response to mechanical tension using the Floyd–Warshal algorithm from the dynamic network structure of the protein.

However, the major roadblock for SMD of Cdh23 is its giant length of EC domain (Cdh23 EC1-27, MW: 320 kD) and lack of structural knowledge of all the EC domains[51]. We, therefore, use a truncated variant of Cdh23, Cdh23 EC1-5 that can form the complete complex with Pcdh15 as in tip-links (EC1-2 of Cdh23 and Pcdh15) and also propose the force-propagation pathways through the noninteracting domains after mechanical perturbations. Among EC1-5 of Cdh23, only EC1-3 is structurally solved (PDB: 5W4T). We thus modeled the EC 4-5 domains using the I-TASSER server[60,61] and aligned EC1-3 and EC 4-5 in PyMOL[62] based on amino acid sequence match. Next, we used the aligned

**Table 1 The kinetic parameters of all unfolding steps of Cdh23 EC1-27 in PAP mode.**

| Extension (nm) | Lifetime (s) (mean ± s.d.) | Transition state distance ($x_\beta$) (nm) (mean ± s.d.) | Energy barrier ($\triangle G^*$) | $F_C (pN) = \triangle G^*/x_\beta$ |
|---|---|---|---|---|
| 6.5 ± 0.5 | 0.038 ± 0.007 | 0.096 ± 0.009 | 17.4 $k_B T$ | 181.2 |
| 16.9 ± 1.4 | 0.027 ± 0.003 | 0.054 ± 0.005 | 17.1 $k_B T$ | 316.6 |
| 24.6 ± 0.6 | 0.052 ± 0.004 | 0.079 ± 0.006 | 17.7 $k_B T$ | 224.0 |
| 33.2 ± 0.9 | 0.061 ± 0.004 | 0.074 ± 0.005 | 17.9 $k_B T$ | 241.9 |

**Table 2 The kinetic parameters of all unfolding steps of Cdh23 EC1-27 in HAP mode.**

| Extension (nm) | Lifetime(s) (mean ± s.d.) | Transition state distance ($x_\beta$) (nm) (mean ± s.d.) | Energy barrier ($\triangle G^*$) | $F_C (pN) = G/x_\beta$ |
|---|---|---|---|---|
| 6.4 ± 0.4 | 0.026 ± 0.004 | 0.086 ± 0.008 | 17.1 $k_B T$ | 198.8 |
| 15.8 ± 0.3 | 0.010 ± 0.003 | 0.071 ± 0.006 | 16.2 $k_B T$ | 228.2 |
| 24.5 ± 0.3 | 0.039 ± 0.003 | 0.108 ± 0.007 | 17.5 $k_B T$ | 162.0 |
| 33.7 ± 0.2 | 0.030 ± 0.005 | 0.088 ± 0.004 | 17.2 $k_B T$ | 195.4 |

structures of Cdh23 domains (EC 1-3 and EC 4-5) and constructed the Cdh23 EC1-5 using template-based homology modeling in SWISS-MODEL[63]. Finally, we minimized the energy of the modeled Cdh23 EC1-5 structure using all-atom Gaussian Accelerated Molecular Dynamics (GAMD) simulations using NAMD (version 2.14) (See Methods). CHARMM36 force field and TIP3P water model are used for the simulations. Finally, the refined structures from GAMD are used in the constant velocity SMD simulations at three relatively slow pulling velocities, 1 Å/ns, 2.5 Å/ns, 5 Å/ns, and with spring of stiffness 7 kcal mol$^{-1}$ Å$^{-2}$. To model the HAP configuration in SMD we anchored the C-terminus of the protein and pulled from N-terminal (Fig. 3a). For PAP configuration, we first obtained the complex structure of Cdh23 EC1-5 and Pcdh15 EC1-2 from the homology modeling of the known heterodimeric structure of Cdh23 EC1-2-Pcdh15 EC1-2 complex (PDB ID: 4AQ8). To model PAP configuration, we used the tip-link complex comprising Cdh23 EC1-5 and Pcdh15 EC1-2, where the C-terminus of Cdh23 EC1-5 was anchored and the complex was pulled from the axially opposite C-terminus end of Pcdh15 EC1-2 (Fig. 3b).

We construct the dynamic network structure of the protein from the correlation of positional fluctuations of residues over a large number of SMD trajectories. We consider SMD trajectories till the domain unfolding. αC atoms of all residues are regarded as nodes in the network structure. Edges in the network are created between nodes if both of them stay within the vicinity of 4.5 Å for 75% of the simulation time. To illustrate the physical meaning of a network structure, we then perform the centrality measurements and deduce the importance of nodes in the network. 'Closeness centrality' denotes the inverse of all possible shortest distances from one node to others[64,65]. A Higher value of closeness centrality of a node indicates shorter distances to all other connected nodes. Thus, the traverse of information (directed mechanical tension) is more effective through a node possessing higher closeness centrality. We measure higher closeness centrality for each residue in HAP than PAP (Fig. 3c), indicating that the transfer efficiency of mechanical force is more elevated in the HAP mode of unfolding. 'Betweenness centrality' of a node denotes the number of times it falls in the shortest distance between two nodes. We estimate higher betweenness centrality of residues in PAP than HAP (Fig. 3d), indicating a more compact, homogeneous, and well-connected residue in PAP than HAP. Overall, the high betweenness centrality and low closeness centrality for PAP suggest a dense intramolecular communication for information passage in PAP than HAP, and thus a better dissipation of information in PAP over HAP.

To identify the web of nodes critical for information transmission, we determine all possible suboptimal paths of force-propagation that are within the 20 nodes from the optimal path. As reported previously, suboptimal paths of force transmission from one protein to the partner in PAP pass through an orthogonal pathway, thus reducing mechanical perturbation on the partner complex[49,66]. Further, we notice a wider spread of suboptimal paths throughout the β-strands and loops in the noninteracting domains of Cdh23 EC1-5 in PAP configuration (Fig. 3f), indicating high efficiency in force-dissipation. Whereas, in HAP, the suboptimal paths instead follow narrow spreading in Cdh23 EC1-5, encompassing fewer nodes. The narrow distribution of the paths limits the propagation of external stimuli through a smaller number of nodes, hence less dissipation of stimuli, making the protein vulnerable toward mechanical force (Fig. 3e).

**PAP is equivalent to spatially distributed multiple-point pulling over a single-point pulling in HAP.** In HAP, the first point of pulling is at the terminal where force transmission occurs from the handle to the anchored protein via a single point. Whereas, in PAP, the force transmission from one protein to the partner depends on the binding strength and the area of the binding interface. The Cdh23-Pcdh15 complex engages in multiple H-bonds and salt-bridge interactions between the interacting EC1-2 domains. Thus, an applied mechanical tension at the C-terminal of Pcdh15 transfers to Cdh23 at multiple points in the binding interface (Fig. 3f, Supplementary Fig. 7). We, therefore, hypothesize that the PAP mode is stereographically a 'spatially distributed multiple-point pulling' of HAP. To test, we have identified the residues on Cdh23 that fall in the suboptimal paths of force transmission from Pcdh15 to Cdh23 with high probability. We then perform SMD simulation by anchoring the C-terminal of Cdh23 and pulling the Arg(71), Gln(137), Ser(142) in independent simulations (Fig. 3g, inset). The SMDs mimic the HAP mode of pulling except for different pulling residues. Subsequently, we determine the suboptimal path of force propagation for individual SMD trajectory and superimpose all paths on Cdh23 E C1-5 (Fig. 3g). As expected, the combined network of three different SMDs from three different residue pulling shows the characteristics of suboptimal paths obtained in PAP mode. Paths are widely distributed throughout β-strands in the individual domain, thus validating our hypothesis that PAP mode of pulling can be attributed to a spatially distributed multipoint pulling scenario.

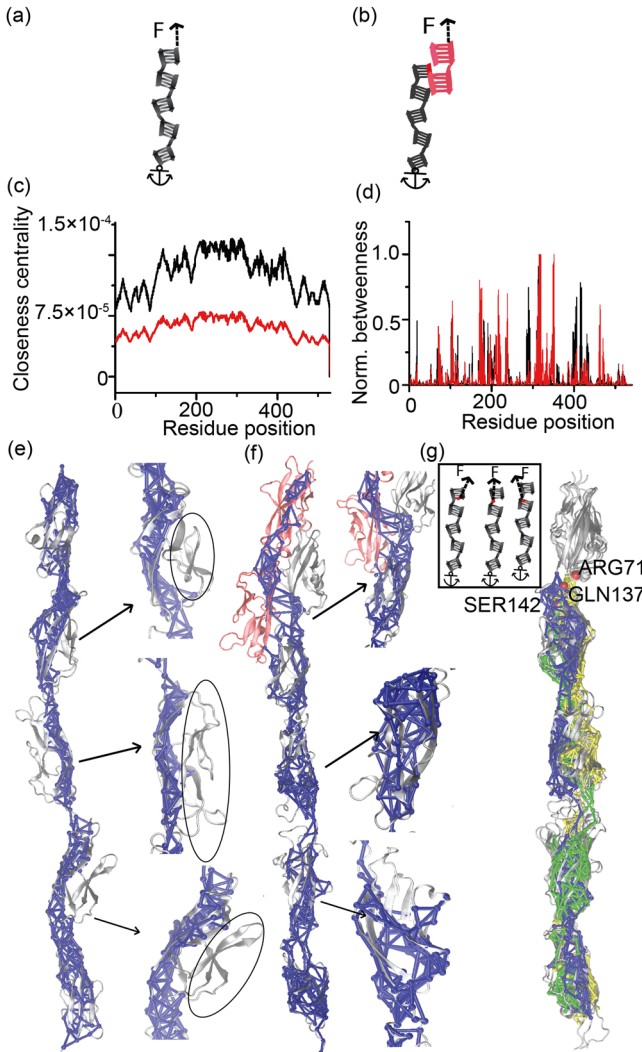

## Discussion

Mechanical force orchestrates many significant biological processes via directly triggering signaling pathways, inducing allostery, altering conformations of bio-macromolecules, and changing ligand-receptor affinities or more. Nature has thus developed various biological force sensors/conveyors (natural spring) to aid the process of force perception in organisms. Some force-sensors bridge the loading point and support point independently and respond to mechanical stimuli directly. Titin protein that defines the elasticity of muscle is the most popular member in this group. Generally, springs in this group perceive a higher magnitude of mechanical stimuli. The other group of force-sensors, prevalent in nature, are protein-protein complexes with an overlapping binding interface. Tip-links, cell-adhesion complexes are typical examples in this group. They perceive a relatively low magnitude of force. While the mechanics of the independent force sensors have extensively been elucidated using state-of-the-art single-molecule manipulation tools, the functional difference of the second type of force-sensors is largely missing. A clear understanding of the functional difference may lead us to identify the evolutionary thirst of nature for both types of force-sensors. Using an AFM-based force-generator, here we decipher the differences in the force-responsive properties of a biological spring protein, Cdh23, between direct pulling (HAP) and indirect pulling via a partner (PAP).

The mechanical response of Cdh23 differs in HAP and PAP. In PAP mode, Cdh23 not only withstands a higher magnitude of force than HAP but also prefers unfolding with the shorter extension of contour lengths. The shorter extension of Cdh23 of ~6.5 nm refers to linker extension. Preference in linker extension over domain unfolding in PAP may facilitate the faster response of tip-links as force-conveyor as well as dissipater. It may be noted that force-induced extension in contour length directly relates to energy-dissipation. Lower the extension, lower is the dissipation, conversely lower energy storage with the folded form. Thus, it is pertinent to say that the protein-complex as force-conveyor may be effective at the low-force stimuli. The potential energy landscape of unfolding of Cdh23 also differs between HAP and PAP; noticeably, the width ($x_\beta$) of the potential barrier is higher in HAP than PAP. Higher resistance to mechanical perturbation and steep potential barrier of protein unfolding in PAP clarifies why unbinding of protein-protein complexes under force in SMFS rarely notice any unfolding prior to dissociation. In contrast, the same force range is sufficient to unfold proteins via direct pulling[67–69].

In-depth analysis of the dynamic network structures of Cdh23 under mechanical tension reveals higher efficacy in the information transformation through the amino acid residues in Cdh23 during HAP than PAP. Mechanical stimulus is the information here. Efficient force transfer makes the amino acid network in Cdh23 less mechano-resistive in HAP. Estimation of force propagation pathways that correlate with the mechnaostability of proteins supports the distinct differences between HAP and PAP. In HAP, the sub-optimal paths of force propagation from the force-loading point to support point are narrowly distributed among Cdh23 domains, thus impeding the load distribution evenly throughout Cdh23, effectively reducing the dissipation efficiency of external mechanical perturbations. In PAP, the force propagation from the force-loading partner to support partner transmits through orthogonal paths, thus limiting the effective work done by the force on support protein instantaneously. Further, the suboptimal paths of force-propagation through Cdh23 are evenly distributed throughout the domains, thus involving a large number of amino acid residues in disseminating the external perturbations and making the complex spring less vulnerable.

**Fig. 3 Dynamic network structures of Cdh23 under tension in HAP and PAP modes. a** Schematic representation of HAP in SMD. Black blocks represent the domains of Cdh23 EC1-5, where the C-terminal is anchored, and the N-terminal is pulled with a spring. **b** Schematic representation of PAP in SMD, depicting Cdh23 EC1-5 in black and Pcdh15 EC1-2 in red. Arrows guide the direction of the force. **c** Closeness centrality obtained for all the residues of Cdh23 EC1-5 in HAP (black) and PAP (red) configurations have been plotted. Residues register a higher closeness centrality value in HAP compared to PAP. **d** Normalized betweenness centrality obtained for all the residues of Cdh23 EC1-5 in HAP (black) and PAP (red) shows higher betweenness values in PAP than HAP for most of the residues. **e** All the sub-optimal force-propagation paths (blue) from N-termini of Cdh23 EC1-5 to C-termini for HAP obtained from a dynamic network analysis of SMD pulling trajectory are overlaid on the ribbon structure of Cdh23 EC1-5 (gray). Here, solid blue tubes denote edges, and blue spheres denote nodes. Arrows guide the zoomed regions of respective EC2, EC3, EC4 domains and the domain regions devoid of any suboptimal path of force propagation are marked with black oval lines. **f** Sub-optimal force-propagation paths for PAP overlaid on the ribbon diagram of complex Cdh23 (EC1-5)-Pcdh15 (EC1-2). Zoomed regions of EC2, EC3, and EC4 highlight the widely distributed sub-optimal paths in all the strands of domains, unlike suboptimal paths in HAP of Cdh23. **g** Inset shows schematics of three independent SMD simulations of Cdh23 EC1-5 pulling from residues R71, Q137, and S142 and anchoring C-termini of Cdh23 EC1-5. Suboptimal paths of force propagation are constructed for all three independent simulations. Suboptimal paths obtained from pulling using R71 (yellow), Q137 (blue), S142 (green) are overlaid on the cartoon representation of Cdh23 EC1-5 (gray).

Why are the suboptimal paths distributed throughout the domains of Cdh23 in PAP? We hypothesized that the simultaneous pulling of Cdh23 from spatially distributed multiple points in PAP is responsible for the difference in mechanoresponse in oppose to terminal pulling in HAP. Anisotropy in the mechanoresponse of proteins with the pulling geometry and directions is known[38,40]. We thus refer PAP as combinations of the anisotropic response of proteins when simultaneously pulled from spatially distributed multiple points, thus effectively at multiple directions. Accordingly, we run three independent SMD simulations on Cdh23 EC1-5 pulling from the three most probable residues that lie at the overlapping interface of the complex and play a seminal role in force transmission from Pcdh15 to Cdh23. Combined suboptimal paths from all three SMD trajectories show that the network is well distributed throughout the domains. So, pulling from one point in the HAP mode concentrates the force on certain paths rather than distributing it throughout the complex, making certain regions extremely vulnerable toward applied force and initiates unfolding of domains in Cdh23. Contrary to HAP, multiple points of force transmission to Cdh23 in PAP diverge the applied force throughout the domains of Cdh23 protein and reduce the effect of applied force on the domains.

## Conclusion

Force application on Cdh23 using its interacting partner Pcdh15 and traditional handle-assisted pulling from terminals (using streptavidin-biotin conjugate handle here) yield variable elastomeric responses of Cdh23 exhibiting stronger mechanical fold architecture in the preceding approach. Stronger mechanical fold architecture is inferred from the higher force-resilience properties in SMFS studies. Higher force-resilience is attributed to the multipoint force application in a partner-assisted pulling leading to distribution of force throughout Cdh23 fold modules. A narrow distribution of force in distinct regions of Cdh23 is noticed in direct pulling using streptavidin-biotin conjugation. Our study may infer that the force-sensors in the inner ear as a heteromeric complex of Cdh23 and Pcdh15 provide extra resilience to input force from a wide range of sound stimuli (5–120 dB in human). Although, the universal applicability of our method requires extensive study with other elastic protein systems in both the force-steering configurations, our findings indicate the importance of measuring the viscoelastic properties of biological springs in their physiological configuration.

## Materials and methods

**Cloning, expression, and purification of mammalian expressed proteins**. We recombinantly modified the N-terminus of Cdh23 EC1-27 construct with Avitag (15 amino acid sequence, PLGGIFEAMKMELRD)[70] and the C-terminus with sort-tag (LPETGSS), GFP, and 6× His-tag, respectively. While GFP was to monitor the expression level, the sort-tag was used to covalently attach the protein onto the surface for force spectroscopy studies. Avitag was used for HAP using streptavidin-biotin. Further, to express the protein in the media, we included a signal peptide sequence before Avitag. All constructs, recombinant Cadh23 EC1-27 and Pcdh15 EC1-2, were cloned in pcDNA3.1 (+) vector for expression. All proteins were expressed in the Expi-CHO expression system (ThermoFisher Scientific). Transfection was done according to the manufacturer's guidelines and then incubated for 7 days. After 7 days, we centrifuged the cells at 2000 rpm for 15 min and collected the media. We dialyzed the media in HEPES buffer at 4 °C and purified the proteins using Ni$^{2+}$-NTA (Qiagen) based affinity chromatography. The composition of the buffer is: 25 mM HEPES, 50 mM KCl, 200 mM NaCl, and 50 μM CaCl$_2$ (pH-7.6)(Hi-Media). After purification, the presence of proteins was confirmed from the SDS-PAGE and western blots.

**Surface modification protocol**. Glass-coverslips were activated using air plasma and subsequently washed with piranha solution for 2 h, followed by a thorough wash with deionized water. Coverslips were then etched using 1 M KOH for 15 min and washed with deionized water by sonicating for 10 min three times. Subsequently, the surfaces were silanized using v/v 2% APTES (3-Aminopropyltriethoxy silane) (Sigma-Aldrich) in 95% acetone and cured at 110 °C for 1 h. The amine exposed surfaces were reacted with Maleimide-PEG-Succinimidyl ester

(Mal-PEG2-NHS) (Sigma-Aldrich) in a base buffer (100 mM NaHCO$_3$, 600 mM K$_2$SO$_4$, pH 8.5) for 4 h. The PEGylated surfaces were subsequently incubated with 100 μM polyglycine peptide, GGGGC, at room temperature (RT) for 7 h for cysteine–maleimide reaction. Polyglycine on coverslip acts as a nucleophile for sortagging. Coverslips were then washed thoroughly with water and stored in a vacuum desiccator prior to protein attachment.

We used less stiff Si$_3$N$_4$ cantilevers (NITRA-TALL from AppNano Inc., USA) for force-spectroscopy studies. After silanization, the cantilevers were treated differently for PAP and HAP. For PAP, the cantilevers were PEGlyated with Maleimide-PEG-Succinimide ester (Mal-PEG2-NHS), and then Polyglycine reaction was performed as described above to attach Pcdh15 EC1-2 covalently.

For HAP, after silanization, cantilevers were PEGylated with NHS-PEG2-NHS followed by incubation with 0.1 mg/mL Streptavidin (Sigma-Aldrich). Finally, cantilevers were washed in a buffer to remove excess streptavidin and stored at 4 °C until use.

**Single-molecule force spectroscopy using AFM**. C-terminus of Cdh23 was immobilized on the polyglycine coated glass coverslip using sortagging reaction in the presence of enzyme Sortase A[50]. After sortagging, biotinylation reaction was performed at the N-terminus of attached Avitag-Cdh23 protein using in-vitro BirA biotinylation protocol[71,72]. For this, we incubated the Cdh23 attached coverslip with mixture of Biotin (50 μM), ATP (10 mM), BirA (1 μM), MgCl$_2$ (10 mM) in low salt SEC buffer (10 mM HEPES, 50 mM KCl, 10 mM NaCl, and 50 μM CaCl$_2$) at RT for 1 h. Then, we washed the coverslip two times with buffer. Similarly, the C-terminus of Pcdh15 EC1-2 was immobilized on Si$_3$N$_4$ cantilevers using sortagging protocol. The Spring constant of the cantilevers was measured following the thermal fluctuation method[73]. After modifying the coverslip and cantilever with proteins, we performed dynamic force-ramp measurements using Atomic Force Microscope (AFM) (Nano wizard 3, JPK Instruments, Germany). We brought the cantilever down at 2000 nm s$^{-1}$, waited for 0.5 s for proteins to interact, and finally retracted at velocities varying from 500, 1000, 2000, 3000, and 5000 nm s$^{-1}$. At each pulling speed, we recorded 10,000 force curves. All experiments were repeated three times with fresh batch of proteins. Analyses of the force-extension curves were performed using already validated home-written MATLAB programs.

For the control experiment, we modified the surface of the coverslip with Cdh23 EC1-27 and pulled the protein with polyglycine-coated cantilevers via non-specific attachments. We obtained an overall 0.4 ± 0.2% of events for each loading rate which is 1/5 times lower than specific pulling either in HAP or PAP. Further, we observed an end-to-end extension of 48.7 ± 1.2 nm for all the events obtained in the control experiment, whereas end-to-end extension for specific pulling peaks at 256.6 ± 27.2 nm in HAP and 186.6 ± 24.4 nm in PAP.

Multiple binding/unbinding is another crucial issue in SMFS. To overcome, several strategies are out in the literature. The density of molecules on surfaces is one such important parameter that is controlled to reduce multiple interactions in SMFS. Interactions with multiple molecules at a single pulling are a common problem in both HAP and PAP. We took a systematic approach to reduce such multi-molecule interactions. We control the density of specific protein molecules by using a mixture of bi-functional and mono-functional PEG at varying ratios. In our previous work (*Biochem J* (2019) 476 (16): 2411–2425), we measured the interaction strength between Cdh23 EC1-2 and Pcdh15 EC1-2 using dynamic force spectroscopy. For Cdh23 EC1-2 vs. Pcdh15 EC1-2, where no unfolding associated unbinding was noticed, 2% bi-functional PEG was doped with mono-functional PEG. We observed more than 97% of force curves with single unbinding features, indicating that 2% surface coverage by proteins is good enough for detecting single unbinding events accurately for proteins with two domains. We then performed unbinding experiments with Cdh23 EC1-10 and Pcdh15 EC1-2 at 2% and 1% surface coverages. Notably, pulling Cdh23 EC1-10 with Pcdh15 EC1-2 undergo unfolding associated unbinding. We observed 41 ± 3% of events featuring unfolding before unbinding irrespective of surface coverage, indicating that the features are majorly contributed from unfolding associated unbinding and not from multiple unbinding. We, thereafter, fixed the surface coverage to 1% and performed experiments with Cdh23 EC1-27 by pulling with Pcdh15 EC1-2. The percentage of force curves featuring multiple unfolding did not increase significantly. However, the number of unfolding per force curves increased with domain numbers. Together with Poisson distributions, these observations indicate that our experimental data are dominated by unfolding associated unbinding and not by multiplex unbinding. However, absolute quantification of the contributions from multiple interactions is impossible in our case.

**Worm-like chain (WLC) model fit**. Unfolding force and change in contour length ($\Delta L_c$) of proteins upon unfolding was measured from the fitting of WLC equation (Eq. 1) of polymer elasticity[53] to the sawtooth force-extension patterns.

$$F(x) = \frac{k_B T}{p} \left[ \frac{1}{4} \left( 1 - \frac{x}{Lc} \right)^{-2} - \frac{1}{4} + \frac{x}{Lc} \right] \tag{1}$$

where $F$ is the unfolding force, $p$ is the persistence length, $x$ is the end-to-end length, and $Lc$ is the contour length of the protein, $k_B$ is the Boltzmann's constant, and $T$ is the temperature. After fitting each peak, unfolding force ($F$) and contour length ($Lc$) is obtained for all different pulling velocities.

Contour length change ($\Delta L_C$) was measured from the difference between two consecutive $Lc$ values of unfolding peaks and plotted as histograms. Most probable $\Delta L_C$ values were estimated from the Gaussian fitting of the histograms. Unfolding forces measured from the WLC fits were plotted as histograms for each pulling velocity. For each histogram, we construct frequency distribution using the Kernel density smoothing function (in MATLAB). For the Kernel density estimation, we determined the bandwidth using the following equation[46,74–76]

$$H = 1.06n^{-1/5}\sigma$$

H: bandwidth, n: number of measurements and $\sigma = \min\{\sigma_x, \text{IQR}/1.34\}$ where $\sigma_x$: standard deviation and IQR is the interquartile range.

Further, LR were estimated directly from the multiplication of spring constant and pulling velocity of the cantilever.

Kinetic parameters such as intrinsic unfolding transition-rate ($k_u^0$) and distance to transition state ($x_\beta$) for unfolding were obtained from the fitting of $F_{mp}$ vs. loading rate plots with Bell–Evans equation[55,56].

$$F(LR) = \left(\frac{k_B T}{x_\beta}\right)\ln\frac{(LR)x_\beta}{k_u^0 k_B T} \quad (2)$$

Activation barrier height ($\triangle G^*$) was estimated from Arrhenius equation:

$$\triangle G^* = -k_B T.\ln\left(k_u^0/A\right) \quad (3)$$

Here, $k_u^0$ is the unfolding transition rate, and A is the Arrhenius frequency factor. For protein dynamics, the value of A is $10^9\,\text{s}^{-1}$ [38,77].

**Molecular dynamics simulation.** Systems for the MD simulations were prepared using QwikMD[58] plugin of VMD[78]. Protein was aligned to the Z-axis and solvated using TIP3P water system in a box maintaining a distance of 15 Å from the edges of the box. Na$^+$ and Cl$^-$ ions were placed at a final concentration of 150 mM randomly in the box. Simulations were performed using NAMD[57] version 2.14 and CHARMM36 force field[79]. After fixing the atom-positions of the protein molecule, the system was minimized for 5000 steps and then 5000 steps of system minimization without any restraints, followed by increasing the system's temperature to 300 K at a rate of 1 K for 600 steps. Equilibration of the system was performed for 5 ns. Then GAMD simulation was performed for 20 ns[80,81]. Noose–Hover method[82,83] was followed to maintain 1 atm pressure, and the Particle mesh Ewald (PME) method[83] was followed to treat the long-range interactions.

For steered molecular dynamics, we used the structure of Cdh23 EC1-5, refined from GAMD. To mimic HAP, the C-terminus of Cdh23 EC1-5 was anchored, and pulling was carried out from N-terminus. In PAP, a complex of Cdh23 EC1-5 and Pcdh15 EC1-2 was used where the C-terminus of Cdh23 EC1-5 was anchored, and pulling was performed from the C-terminus of Pcdh15 EC1-2. In the multipoint pulling scenario, pulling was performed in three independent simulations by anchoring the C-terminus of Cdh23 EC1-5 and pulling using Arg71, Ser142, and Gln137 residues of Cdh23 EC1-5. All the SMD simulations were carried out at three different pulling velocities of 1 Å/ns, 2.5 Å/ns, and 5 Å/ns.

**Dynamic network analysis.** Dynamic network analysis has been carried out using NetworkView, plugin of VMD[59] and other associated tools. For the network construction, all the α-C's were considered as nodes. Edges were created between two nodes if they lie in the vicinity of 4.5 Å of a heavy atom for 75% of the simulation time. Neighboring alpha carbon atoms were not considered for the analysis procedure as it will lead to trivial paths. Edges were weighted using correlation coefficient calculated from correlation matrix using carma[84]. Suboptimal paths were determined using subopt script of VMD, which uses the Floyd–Warshal algorithm[85]. For HAP, we estimated the suboptimal paths between the N-terminus and C-terminus of Cdh23 EC1-5. For PAP, the estimation of suboptimal paths started from the C-terminus of Pcdh15 EC1-2 and extended till the C-terminus of Cdh23 EC1-5.

**Statistics and reproducibility.** We have performed all the experiments in triplicates with fresh batch of proteins by using new set of coverslip and cantilevers. Erros are estimated as standard error from experiment replicates and standard error of fitting as applicable.

**Associated content.** Supplementary figures and tables are attached separately in supplementary document.

**Reporting summary.** Further information on research design is available in the Nature Research Reporting Summary linked to this article.

## Data availability
The source data underlying the graphs are available as Supplementary Data 1. Any remaining information along with original force curves are available from corresponding author upon reasonable request.

## Code availability
All home-written MATLAB codes were used for the analysis and codes are available at a Github repository https://github.com/Singlemoleculelab-IISERM/Arora-et-al-Communicaton-Biology[86].

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

## Acknowledgements

This work was supported by the Wellcome Trust/ DBT Indian Alliance fellowship [grant number: IA/I/15/1/501817] awarded to SR. SR acknowledges the financial support provided by The Wellcome Trust/DBT Intermediate fellowship by Indian Alliance and Indian Institute of Science Education and Research Mohali, India (IISERM). We thank Professor Raj Ladher, National Centre for Biological Science, India for providing Cdh23 and Pcdh15 mammalian constructs. NA is thankful to CSIR-India for providing fellowship. JPH is thankful to IISERM for the financial support.

## Author contributions

S.R. conceived the idea. J.P.H., N.A., and S.R. designed all the experiments and analyzed the data. N.A. and J.P.H. expressed and purified the proteins. J.P.H. and N.A. made the figures. J.P.H. and S.R. wrote the paper. N.A., S.R. and J.P.H. edited the paper.

## Competing interests

The authors declare no competing interests.
