## [Peer Review File · Communications Biology]

Reviewers' comments:

Reviewer #1 (Remarks to the Author):

Rajkshit and coauthors study the force dynamics of cadherin-23 using SMFS and network analysis generated from SMD. Their data indicates that the hand-shake geometry drives the force-response of cadherin-23 through a different potential-energy landscape than direct pulling, which is an interesting result that will be of interest to the bio-SMFS community. I have a few comments that the authors may consider in revising their manuscript:

1. The paper is littered with typos and poor grammar, which makes the paper difficult to read and in a few cases, makes it difficult to understand exactly what the authors are trying to say; I strongly recommend proofreading or professional copy editing.
2. For the simulations, more details would be useful on the parameter choices made for the pulling force and the construction/definition of the network edges and nodes. The main text would be more balanced and useful by including some discussion of this, in the context of the strength and limitations of the SMD/network model to compare with experimental SMFS given the inherently "too fast" simulations of forced unfolding.
3. The x-axes in Fig 2k,l need scale bars.

Reviewer #2 (Remarks to the Author):

The manuscript "Anisotropy in Mechanical Unfolding of Protein upon Partner-Assisted Pulling and Handle-Assisted Pulling" by Arora et al. details experimental and simulation results on the pulling of the 27 extracellular domains of protein cadherin-23 attached through a direct end-to-end linkage or through binding one end to the first 2 extracellular domains of protocadherin-15. The Authors show a difference in the mechanical unfolding due to linkage type and interpret the results through simulation.

Overall, I think this is a good manuscript that could be of interest to the general community. I believe the Authors should address a few points that I have included below.

Comments:

1. You plot in figure 2, and use in your analysis, the force of unfolding vs loading rate in pN/s. The loading rate was found by multiplying the rate of retraction (in nm/s) with the cantilever's spring constant. Since the sample is a non-linear spring and the two different connection types may change the characteristics of this spring, how do the nominal loading rates at unfolding compare to the experimental values? The experimental values can be found through the slope of the extension vs force curves at breakage times the retraction speed.
2. You used the entire 27 domain long Cdh23. How many unfolding events did you see, on average, during pulling?
3. How do you know that any of these force peaks (specifically the ones with shorter delta-Lc) are not due to release of domains stuck to the glass non-specifically and not due to unfolding? In either case, there would be an increase in the length due to the added domain. What controls were performed to test for non-specific attachment?
4. What was done to test or correct for multiple Cdh23 attachments to the cantilever or Pcdh15-coated cantilever? Some of the force distributions (see the delta Lc 6.5 for PAP mode) has very broad distributions that may have several peaks. Could the higher forces in PAP mode be an indicator of Pcdh15 binding to more than one Cdh23?
5. Buffers for the experiments contained 50 μ M CaCl₂. Since Calcium affects Cdh23, please comment on whether this is a high or low value for Cdh23 and whether you can predict if the linkers are bound to Ca or not at this concentration.

Reviewer #3 (Remarks to the Author):

The present manuscript evaluates the mechanoresponse of protein complexes using single molecule AFM studies. The authors have studied cadherin-23 as the model protein in two different pulling assays : hand (HAP) and partner (PAP) assisted pulling modes. The data shows higher unfolding forces and lower distance to transition state for the unfolding transitions observed in the PAP mode than in HAP. Using dynamic network structure analysis of Cadherin-23 under force, the authors have postulated that the PAP mode provides spatially distributed multiple-point pulling due to the Pcdh15 - Cdh23 interface/interaction. However, the manuscript is in a preliminary stage in terms of the experiments performed, data analysis methods used and due to not clear writing/explanations. Thus at this stage I cannot recommend it for publications. The main issues are as follows:

1. In Fig 1 (c-e) (d-f), it is not clear what is plotted on the x axis? The label shows contour lengths L_c but it should be the end to end distance of the proteins ?. Also why Fig 1 e has more noise than 1 c HAP data ? Since the loading rate is higher in the pulling trace of 1 e, it is expected that the noise should be lower not higher ? This raises concerns about the calibration of the set up and the response of the cantilever.

2. In Fig 1f , PAP mode, the extension values are showing 250 nm as the fully unfolded state while in 1 d the fully unfolded state is above 300 nm. The authors have not explained how the change in loading rates results in these differences. The unfolding of the Pcdh15 domains can also contribute in the unfolding transitions observed in the PAP mode. The authors have not provided controls to check this. Moreover, both HAP and PAP mode traces are showing heterogeneity in the number of unfolding transitions for different loading rates. Thus it is not clear what is the native state unfolding pattern of the proteins in the two modes. It will be best if for each loading rate more examples of the traces are provided to get a better understanding of the unfolding pattern.

3. The authors have reported 4 main unfolding transitions identified in the two modes. The histograms of the contour length (Fig 1 g-h) changes are plotted by grouping the data from all the force extension curves. It is not explained at which loading rate this analysis is done. Also, this method is correct if the unfolding patterns of the curves are similar for each mode. However as mentioned in point 2 above the traces looks heterogeneous in their unfolding pathways and the data categorization should be done based on each class of unfolding pattern. This will also help in the proper understanding of which domains are contributing in the measured transitions. It is important to know which of these transitions corresponds to the EC1-2 of Cdh23 and Pcdh15 interactions and why all the transitions shows higher unfolding forces.

4. I figure 2 (a-d), the unfolding force distributions were fitted with a gaussian distribution. It is not clear why the force distributions were considered gaussian. Many histograms are showing poor fitting with offset values not reported. The authors should use the correct probability distribution of forces, see Schlierf et al. PNAS 2004 and Woodside et al. Annual Review of biophysics 2014.

5. In fig 2 i and k, the energy landscapes are drawn showing the PAP and HAP assays with the same native state energy level. Since the PAP mode is a complex (Pcdh15 - Cdh23) how is this possible. The authors should provide bulk experiments (thermal/chemical denaturation) to get an estimate of the stability of the two assays used. The energy landscapes are shown at zero force but on the x axis extension values are plotted. The extension of the proteins is a function of force so how it is calculated at zero force. the energy diagrams should be plotted with contour length changes as the reaction coordinate. Note that the experiments do not show any refolding thus the barrier from unfolded to transition state should not be shown. Also to calculate the barrier heights the authors have reported the pre exponential factor (A) = 10^9 s^{-1} . Why this value is chosen?

6. The barrier height for the 24.5 nm transition (Fig 2 i) is much higher for HAP mode than what is shown for PAP mode in fig 2 k. However, in Table 1 no difference in the barrier height for this transition is reported.

7. The authors have used dynamic network structure analysis and have concluded a spatially

distributed multiple pulling geometry in the PAP mode that results in force dissipation. Although the use of this method is appreciated however it is not directly explaining the AFM results. The best approach would be to directly measure EC1-2 of Cdh23 and Pcdh15 interaction strength using AFM which is a key parameter in the analysis. This will shed light on the binding strength at the interface of the proteins and negate any possible contribution of the Pcdh15 in the unfolding transitions observed in fig 1.

Reviewer 1:

1. The paper is littered with typos and poor grammar, which makes the paper difficult to read and in a few cases, makes it difficult to understand exactly what the authors are trying to say; I strongly recommend proofreading or professional copy editing.

Reply: *We have extensively worked on the language of the draft and edited it appropriately. We marked the changes in red, including changes related to rewriting of sentences or changing the monotonous language without changing the meaning of the statements,*

Further, we have verified the grammar using the professional version of Grammarly. I find the current version is easier to read. We sincerely thank the reviewer for pointing this out.

2. For the simulations, more details would be useful on the parameter choices made for the pulling force and the construction/definition of the network edges and nodes. The main text would be more balanced and useful by including some discussion of this, in the context of the strength and limitations of the SMD/network model to compare with experimental SMFS given the inherently "too fast" simulations of forced unfolding.

Reply: *In agreement with the reviewer, we have now included appropriate definitions and justifications on simulation parameters and network parameters (including nodes and edges) in the main manuscript. The inclusions are marked in red in the updated version, named "Marked Draft". The corresponding method section contains all the details of the simulations.*

Indeed, the pulling velocities in experimental SMFS are reportedly slower in comparison to in-silico pulling measurements. In silico unfolding of proteins require an exceptionally elongated box to accommodate the unfolded protein form. SMDs in explicit solvent thus make the simulation runs expensive and time-consuming. To compensate, significantly faster-pulling velocities are often executed in SMDs. Our objective here was to map the force-propagation through a folded protein, and we could run the SMDs at much slower velocities than commonly used, closer to experiments. We used a maximum pulling velocity of 10^3 nm/s in experiments, whereas 10^8 nm/s in silico.

Further, to verify how differences in pulling velocities alter force-propagation pathways and force-mediated unfolding pathways, we ran SMDs in three different velocities. We have observed a similar network in all three velocities. So, we safely assumed that the network at the experimental velocities would be similar.

3. The x-axes in Fig 2k,l need scale bars.

Reply: *Figures 2k and 2l are only schematically representing the differences in the potential energy landscapes of protein unfolding in HAP and PAP as obtained from the SMFS experiments. We prefer to keep it qualitative.*

Reviewer 2:

1. You plot in figure 2, and use in your analysis, the force of unfolding vs loading rate in pN/s. The loading rate was found by multiplying the rate of retraction (in nm/s) with the cantilevers spring constant. Since the sample is a non-linear spring and the two different connection types may change the characteristics of this spring, how do the nominal loading rates at unfolding

compare to the experimental values? The experimental values can be found through the slope of the extension vs force curves at breakage times the retraction speed.

Reply: As suggested, we have determined the loading rates from all the force curves according to the slope-based method for both HAP and PAP mode. Next, we have plotted the mean unfolding forces for each type of extension with loading rates to determine the kinetic parameters of unfolding. As shown in the Table below, quantitatively, the newly estimated lifetime and the distance to the transition state differ slightly from earlier calculated values. However, the lifetimes of unfolding in HAP are lower than the corresponding unfoldings in PAP as obtained from the earlier method. The distances to the transition state for all the unfoldings are higher in HAP than PAP. So, the different methods employed here for estimating the loading rates do not affect the overall outcome/result of our work. We have therefore included this additional method in the SI (**Figure S6**).

Figure S6: Estimating kinetic parameters of unfoldings from experimentally estimated loading rates: (a) Histograms of loading rate calculated from the slope of the linear portion of force-extension curves obtained in HAP of Cdh23 EC1-27. (b) Histograms of loading rate calculated from the slope of the linear part of force-extension curves obtained in PAP of Cdh23 EC1-27. Loading rates were estimated by multiplying the slope of the curves with the corresponding pulling velocities. Histograms are fit to the Gaussian peak function. (c-f) Most-probable unfolding forces are plotted with the experimentally determined loading rates for HAP and PAP of Cdh23 EC1-27 for different extension types of (c) ~6.5 nm, (d) ~16 nm, (e) ~24 nm, and (f) ~33 nm, respectively.

SI Table 1: (a) The kinetic parameters of all unfolding steps of Cdh23 EC1-27 in HAP mode. (b) The kinetic parameters of all unfolding steps of Cdh23 EC1-27 in PAP mode. The mean±s.d. are estimated from n=3 independent experiments.

(a)

Extension (nm)	Lifetime(s) (mean±s.d.)	Transition state distance (x_β) (nm) (mean±s.d.)	Energy barrier (ΔG^*)	F_c (pN) = $\Delta G^*/x_\beta$
6.4±0.4	0.026±0.004	0.073±0.002	17.1 $k_B T$	234.2
15.8±0.3	0.015±0.005	0.064±0.003	16.5 $k_B T$	257.8
24.5±0.3	0.018±0.007	0.084±0.007	17.3 $k_B T$	198.8
33.7±0.2	0.028±0.005	0.064±0.003	17.6 $k_B T$	267.1

(b)

Extension (nm)	Lifetime (s) (mean±s.d.)	Transition state distance (x_β) (nm)(mean±s.d.)	Energy barrier (ΔG^*)	F_c (pN) = $\Delta G^*/x_\beta$
6.5±0.5	0.0419±0.008	0.095±0.002	17.5 $k_B T$	184.2
16.9±1.4	0.0413±0.004	0.042±0.003	17.5 $k_B T$	416.6
24.6±0.6	0.0308±0.007	0.046±0.004	16.7 $k_B T$	373.9
33.2±0.9	0.0468±0.005	0.044±0.003	17.1 $k_B T$	401.2

2. You used the entire 27 domains long Cdh23. How many unfolding events did you see, on average, during pulling?

Reply: Though we noticed varying numbers of unfoldings per force curves in HAP and PAP and at different loading rates, the average number of unfoldings per force curve ranges between 6 – 9 for both the pulling configurations. We have plotted the average numbers of unfolding per force curve with pulling velocity for HAP (left panel) and PAP (right panel) for further clarity. We have included this Figure in the SI (**Figure S3**).

Figure S3: Unfolding numbers per force curves. (a) The average number of unfoldings of Cdh23 EC1-27 per force curve is shown for HAP in a scattered plot. Error bars denote the standard error of the mean (SEM), where n ranges between 40-70. (b) An average number of unfoldings of Cdh23 EC1-27 per force curve is shown for PAP in a scattered plot. Error bars denote the standard error of the mean (SEM), where n ranges between 40-70.

3. How do you know that any of these force peaks (specifically the ones with shorter delta-Lc) are not due to release of domains stuck to the glass non-specifically and not due to unfolding? In either case, there would be an increase in the length due to the added domain. What controls were performed to test for non-specific attachment?

Reply: This is an excellent remark indeed. Contributions from non-specific attachments are a serious concern in SMFS, and thus quantifying the contributions from non-specific attachments is routinely done. We, too, had considered the contribution from non-specific measurements extensively in our experiment by various means. We have included the following paragraph in the Method section to avoid any further confusion.

Firstly, we modified the surface of the coverslip with Cdh23 EC1-27 and pulled the protein with polyglycine coated cantilevers via non-specific attachments. We obtained an overall $0.4 \pm 0.2\%$ of events for each loading rate which is 1/5 times lower than specific pulling either in HAP or PAP. Further, we observed an end-to-end extension of 48.7 ± 1.2 nm for all the events obtained in the control experiment, whereas end-to-end extension for specific pulling peaks at 256.6 ± 27.2 nm in HAP and 186.6 ± 24.4 nm in PAP.

Secondly, constructs possessing fewer domains should have a lower probability of non-specific attachments to the surface. However, we observed a similar contour length increment in the pulling of Cdh23 EC1-5, Cdh23 EC1-10, and Cdh23 EC1-27. We didn't observe any unfolding of domains during the interaction studies of Cdh23 EC1-2 and Pcdh15 EC1-2 using SMFS.

4. What was done to test or correct for multiple Cdh23 attachments to the cantilever or Pcdh15-coated cantilever? Some of the force distributions (see the delta Lc 6.5 for PAP mode) has very broad distributions that may have several peaks. Could the higher forces in PAP mode be an indicator of Pcdh15 binding to more than one Cdh23?

Reply: This is another crucial issue in SMFS. To overcome, several strategies are out in the literature. The density of molecules on surfaces is one such important parameter that is controlled to reduce multiple interactions in SMFS. Interactions with multiple molecules at a single pulling are a common problem in both HAP and PAP. We took a systematic approach to reduce such multi-molecule interactions. We control the density of specific protein molecules by using a mixture of bi-functional and monofunctional PEG at varying ratios. In our previous work (*Biochem J* (2019) 476 (16): 2411–2425), we have measured the interaction strength between Cdh23 EC1-2 and Pcdh15 EC1-2 using dynamic force spectroscopy. For Cdh23 EC1-2 vs Pcdh15 EC1-2, where no unfolding associated unbinding was noticed, 2% bi-functional PEG was doped with monofunctional PEG. We observed more than 97% of force curves with single unbinding features, indicating that 2% surface coverage by proteins is good enough for detecting single unbinding events accurately for proteins with two domains. We then performed unbinding experiments with Cdh23 EC1-10 and Pcdh15 EC1-2 at 2% and 1% surface coverages. Notably, pulling Cdh23 EC1-10 with Pcdh15 EC1-2 undergo unfolding associated unbinding. We observed 41 ± 3 % of events featuring unfolding before unbinding irrespective of surface coverage, indicating that the features are majorly contributed from unfolding associated unbinding and not from multiple unbinding. We, thereafter, fixed the surface coverage to 1% and performed experiments with Cdh23 EC1-27 by pulling with Pcdh15 EC1-2. The percentage of force curves featuring multiple unfolding did not increase significantly. However, the number of unfolding per force curves increased with domain numbers. Together with Poisson distributions, these observations indicate that our experimental data is dominated by unfolding associated unbinding and not by multiplex unbinding. Though absolute quantification of the contributions from multiple interactions is impossible in our case.

5. Buffers for the experiments contained 50 μM CaCl_2 . Since Calcium affects Cdh23, please comment on whether this is a high or low value for Cdh23 and whether you can predict if the linkers are bound to Ca or not at this concentration.

Reply: 20-60 μM of Ca^{2+} ions is found in the inner-ear endolymph, whereas the dissociation constants for calcium ions are estimated in the range of 40-100 μM . This indicates that at 50 μM of Ca^{2+} ions, all linkers may not be saturated or bound with calcium ions.

Reviewer 3

The present manuscript evaluates the mechanoresponse of protein complexes using single molecule AFM studies. The authors have studied cadherin-23 as the model protein in two different pulling assays, handle (HAP) and partner (PAP) assisted pulling modes. The data shows higher unfolding forces and lower distance to transition state for the unfolding transitions observed in the PAP mode than in HAP. Using dynamic network structure analysis of Cadherin-23 under force, the authors have postulated that the PAP mode provides spatially distributed multiple-point pulling due to the Pcdh15 - Cdh23 interface/interaction. However, the manuscript is in a preliminary stage in terms of the experiments performed, data analysis methods used and due to not clear writing/explanations. Thus at this stage I cannot recommend it for publications. The main issues are as follows:

1. In Fig 1 (c-e) (d-f), it is not clear what is plotted on the x axis? The label shows contour lengths L_c but it should be the end to end distance of the proteins? Also why Fig 1 e has more

noise than 1 c HAP data? Since the loading rate is higher in the pulling trace of 1 e, it is expected that the noise should be lower not higher? This raises concerns about the calibration of the set up and the response of the cantilever.

Reply: (i) I sincerely thank the reviewer for pointing this out. We have modified the axis label to 'Tip-surface distance(nm)'.

All cantilevers are individually calibrated after all chemical modifications during measurements. We measure the power spectra of the cantilevers in between experiments and maintain a constant sum throughout the experiment time. Figure 1e does show relatively higher noise. After the concern raised by the reviewer, we re-checked the force curves, and we noticed such noisy data is there only for a small number of events at the corresponding loading rate. We have no conclusive answer to why such noise appeared in the measurements as these measurements are run automatically for several hours. I can only hypothesize that the contributions could be from electrical noise. We would refute the concerns related to calibration and the response of cantilevers. Further, if the reviewer and editor agree, we can replace this data with better representative curves and show all the curves overlaid in the SI.

2. In Fig 1f, PAP mode, the extension values are showing 250 nm as the fully unfolded state while in 1 d the fully unfolded state is above 300 nm. The authors have not explained how the change in loading rates results in these differences. The unfolding of the Pcdh15 domains can also contribute in the unfolding transitions observed in the PAP mode. The authors have not provided controls to check this. Moreover, both HAP and PAP mode traces are showing heterogeneity in the number of unfolding transitions for different loading rates. Thus it is not clear what is the native state unfolding pattern of the proteins in the two modes. It will be best if for each loading rate more examples of the traces are provided to get a better understanding of the unfolding pattern.

Reply: Figure 1c-f are the representative force curves for HAP and PAP. We observed an average extension of 256.6 ± 27.2 nm in HAP and 186.6 ± 24.4 nm in PAP, independent of loading rates. The total extension in any force curves is a combination of the number of domain unfolding, linker unfolding, and the entropic extension. While the domain-unfolding and linker-unfolding may produce narrowly distributed extensions in the unfolding events, the entropic extension often varies widely. The entropic extensions also depend on surface modifications. Therefore, the total extensions are not straightforward to explain and hardly reported in the literature, even for polyproteins with repeat domains.

We have plotted here the distributions of the total extension for HAP and PAP at each loading rate.

Further, we disagree with the reviewer on measuring the extension of Pcdh15 EC1-2 during unbinding of the complex. Pcdh15 EC1-2 does not unfold prior to unbinding. Previously while quantifying the tip-link complex strength, we performed experiments between Cdh23 EC1-2 and Pcdh15 EC1-2 and did not notice any unfolding associated unbinding (Biochem J (2019) 476 (16): 2411–2425). So, it is unlikely that the two outermost domains undergo any unfolding in PAP before unbinding.

Figure 1. (a) Histograms of the total extensions obtained in each unfolding force-curve for each pulling velocity during HAP of Cdh23 EC1-27. (b) Histograms of the total extensions obtained in each unfolding force-curves at each pulling velocity during PAP of Cdh23 EC1-27.

3. The authors have reported 4 main unfolding transitions identified in the two modes. The histograms of the contour length (Fig 1 g-h) changes are plotted by grouping the data from all the force extension curves. It is not explained at which loading rate this analysis is done. Also, this method is correct if the unfolding patterns of the curves are similar for each mode. However as mentioned in point 2 above the traces looks heterogeneous in their unfolding pathways and the data categorization should be done based on each class of unfolding pattern. This will also help in the proper understanding of which domains are contributing in the measured transitions. It is important to know which of these transitions corresponds to the EC1-2 of Cdh23 and Pcdh15 interactions and why all the transitions shows higher unfolding forces.

Reply: Histograms of the contour length change has been plotted for the data combining all loading rates, 838 unfolding events for HAP and 946 events for PAP. The same is stated in the Figure legend. Since it was not enough for the reviewer to understand, we have included the following statement in the figure description.

“The data is combined for all loading rates.” The change is marked in red in the edited version.

We have categorized the unfolding force curves according to each contour length gain in Figure 2. The distinct analyses for each contour length gain enabled us to determine the thermodynamics parameters for each unfolding type though we specifically have no knowledge yet on their origin.

Both Figures 1 and 2 exclusively describe the unfolding features of Cdh23, and do not infer any information related to the unbinding of Cdh23-Pcdh15 complexes. We have not considered the last peak forces in the analyses. The very last peak in all the unfolding-unbinding force-distance stretching curves here (either HAP or PAP) describes the unbinding of complex, and we judiciously did not include that data here.

4. I figure 2 (a-d), the unfolding force distributions were fitted with a Gaussian distribution. It is not clear why the force distributions were considered Gaussian. Many histograms are showing poor fitting with offset values not reported. The authors should use the correct probability distribution of forces, see Schlierf et al. PNAS 2004 and Woodside et al. Annual Review of biophysics 2014.

Reply: *The fitting values, including offset, error in offset, chi-square, and adjacent Chi-square values, are mentioned in the following Tables. The purpose of the Gaussian fit here is to obtain the peak maxima and for a visual guide to the eyes. We have been conservative here to fit to different distribution probabilities and infer more information from the distributions. Since the protein unfolding here follows heterogeneity and features varieties in the unfolding lengths, we obtained a limited number of events for each case, and thus, the distributions appear skewed.*

Table 1: Gaussian fitting parameters obtained for each force-histograms for contour length change of ~6.5 nm

HAP	Pulling velocity	Offset(Y_0)	Error	Reduced Chi-Sqr
	500	0.10	0.04	0.023
	1000	0.08	0.04	0.032
	2000	0.04	0.02	0.026
	3000	0.03	0.02	0.016
	5000	0.06	0.04	0.049
PAP				
	500	0.04	0.02	0.025
	1000	0.19	0.03	0.04
	2000	0.22	0.03	0.017
	3000	0.06	0.01	0.01
	5000	0.10	0.03	0.02

Table 2: Gaussian fitting parameters obtained for each force-histograms for contour length change of ~16 nm

HAP	Pulling velocity	Offset(Y_0)	Error	Reduced Chi-Sqr
	500	0.09	0.02	0.017
	1000	0.01	0.02	0.011
	2000	0.05	0.03	0.012
	3000	0.07	0.04	0.031
	5000	0.02	0.01	0.022
PAP				
	500	0.02	0.08	0.035
	1000	0.05	0.03	0.03
	2000	0.08	0.04	0.036
	3000	0.03	0.03	0.013
	5000	0.04	0.02	0.017

Table 3: Gaussian fitting parameters obtained for each force-histograms for contour length change of ~24 nm

HAP	Pulling velocity	Offset(Y_0)	Error	Reduced Chi-Sqr
	500	0.05	0.009	0.021
	1000	0.04	0.01	0.024
	2000	0.02	0.02	0.02
	3000	0.04	0.03	0.02
	5000	0.09	0.05	0.025
PAP				
	500	0.10	0.08	0.05
	1000	0.03	0.04	0.022
	2000	0.08	0.03	0.029
	3000	0.08	0.07	0.084
	5000	0.02	0.01	0.011

Table 4: Gaussian fitting parameters obtained for each force-histograms for contour length change of ~33 nm

HAP	Pulling velocity	Offset(Y_0)	Error	Reduced Chi-Square
	500	0.04	0.02	0.016
	1000	0.17	0.05	0.022
	2000	0.12	0.03	0.023
	3000	0.07	0.04	0.031
	5000	0.08	0.03	0.026
PAP				
	500	0.05	0.009	0.026
	1000	0.05	0.03	0.027
	2000	0.02	0.02	0.011
	3000	0.02	0.007	0.013
	5000	0.02	0.01	0.013

5. In fig 2 i and k, the energy landscapes are drawn showing the PAP and HAP assays with the same native state energy level. Since the PAP mode is a complex (Pcdh15 - Cdh23) how is this possible. The authors should provide bulk experiments (thermal/chemical denaturation) to get an estimate of the stability of the two assays used. The energy landscapes are shown at zero force but on the x axis extension values are plotted. The extension of the proteins is a function of force so how it is calculated at zero force. the energy diagrams should be plotted with contour length changes as the reaction coordinate. Note that the experiments do not show any refolding thus the barrier from unfolded to transition state should not be shown. Also to calculate the barrier heights the authors have reported the pre exponential factor (A) = 10^9 s⁻¹. Why this value is chosen?

Reply: The potential energy diagrams in Figures 2 k and l are schematically representing the kinetics of the four distinct unfolding features of Cdh23 alone while pulling in HAP and PAP, respectively. Complexes here are just the springs.

Indeed, the native state energy of the complexes in HAP and PAP are different. However, with the significantly smaller overlapping surface area of the complexes w.r.t. the long length of Ch23 EC1-27 itself, it is safe to assume that the native state energy of the complexes will be dominated by Cdh23 alone. In that case, the energy states of the two complexes in HAP and PAP are negligibly different. Moreover, we have not equalized the native state energies here. HAP and PAP complexes are drawn in two different figures 2k and l, respectively, where the Y axes are unmarked. With a lack of knowledge on the energies of the four unfolded configurations, we merged all four unfolded conformations to a single energy state for both HAP and PAP.

X-axes of Figures 2k and l are modified to 'Reaction Coordinate.'

We have mentioned in Figure legends that the diagrams only schematically represent the **unfolding** kinetics.

For protein dynamics, the pre-exponential factor (A) is chosen to 10^9 s^{-1} . Please see the following references. The references are mentioned in the main manuscript as well.

(a) The speed limit for protein folding measured by triplet–triplet energy transfer- Bieri et al. Proc. Natl. Acad. Sci. USA Vol. 96,9597–9601,1999.

(b) Anisotropic deformation response of single protein molecules-Dietz et al., Proc. Natl. Acad. Sci. USA Vol 103, 12724–12728, 2006.

6. The barrier height for the 24.5 nm transition (Fig 2 i) is much higher for HAP mode than what is shown for PAP mode in fig 2 k. However, in Table 1 no difference in the barrier height for this transition is reported.

Reply: This is a mistake that happened during the drawing of the schematics of the energy diagram. We have rectified it and put the new energy diagram. Thanks to the reviewer for pointing this out.

7. The authors have used dynamic network structure analysis and have concluded a spatially distributed multiple pulling geometry in the PAP mode that results in force dissipation. Although the use of this method is appreciated however it is not directly explaining the AFM results. The best approach would be to directly measure EC1-2 of Cdh23 and Pcdh15 interaction strength using AFM which is a key parameter in the analysis. This will shed light on the binding strength at the interface of the proteins and negate any possible contribution of the Pcdh15 in the unfolding transitions observed in fig 1.

Reply: We have measured the strength of the EC1-2 Cdh23 and Pcdh15 interaction earlier using AFM-based SMFS (Biochem J (2019) 476 (16): 2411–2425). We found the heteromeric interaction to be relatively stable, with an off-rate of $4.5 \times 10^{-3} \pm 4.9 \times 10^{-5} \text{ s}^{-1}$ and distance to the transition state of $0.18 \pm 0.03 \text{ nm}$. So, the binding strength of the heteromer is relatively high. Also, during the measurement of the heteromeric interactions between Cdh23 EC1-2 vs.

Pcdh15 EC1-2, we didn't observe any signature of unfoldings of proteins. Thus, none of the features in Figures 1 and 2 has contributions of Pcdh15 EC1-2 unfolding. Further, the effective contribution of domain stability does not propagate beyond the interacting domains (Binding-induced stabilization measured on the same molecular protein substrate using single-molecule magnetic tweezers and heterocovalent attachments, Dahal et al., J. Phys. Chem. B 2020, 124, 3283-3290).

Reviewers' comments:

Reviewer #1 (Remarks to the Author):

The authors have addressed all my concerns and answered all comments including appropriate changes to the main text and SI. I now find the paper suitable for publication in Comms Bio.

Reviewer #2 (Remarks to the Author):

I believe the Authors have addressed all of my concerns from the original submission and have incorporated the necessary changes into the manuscript. I have no further concerns.

Reviewer #3 (Remarks to the Author):

I am still not convinced with the responses of the authors. Though the writing is improved, the main issues regarding the performed experiments and data analysis methods used are still not explained correctly.

1) As the authors have mentioned that the experimental traces Fig 1 e is due to some noise in the set up, it is not clear then why these traces were used in the analysis. The authors need to first optimize the set up to reduce the noise as it is a must for publishing experimental data. Also, why the data is noisy at a particular loading rate? In my opinion more experiments are needed to get low noise traces in the same condition and then analysis should be done.

2) It is still not clear why gaussian fitting of force distributions is done. The authors have mentioned that they have done it in a qualitative way to get the force peaks however it is not the case. The force values are used to extract other parameters of energy landscape for which correct distribution functions should be used. As the authors have mentioned the heterogeneity in the traces resulted in poor statistics then why energy landscape parameters were quantified. For single molecule measurements statistical significance is must and thus the manuscript looks weak.

3) If the binding strength of EC1-2 of Cdh23 and Pcdh15 is high, why the authors have not reported the forces required to rupture the interactions. These values should be mentioned in the text so that a direct comparison can be made with the unfolding forces of the domains.

1) As the authors have mentioned that the experimental traces Fig 1 e is due to some noise in the set up, it is not clear then why these traces were used in the analysis. The authors need to first optimize the set up to reduce the noise as it is a must for publishing experimental data. Also, why the data is noisy at a particular loading rate? In my opinion more experiments are needed to get low noise traces in the same condition and then analysis should be done.

Reply: As mentioned previously, we noticed such noisy data only for a small number of events. To highlight, we estimated the noise in the force curves as a measure of mean absolute deviation (MAD) from the tail region of each force-curve after unbinding (highlighted in red box) and plotted as a histogram. We notice that the noise distribution is very sharp, with a peak-maxima at 6.7 ± 0.1 pN (**Figure Xa**). A negligible fraction of force curves discretely shows higher noise. Erroneously, we used one such noisy force-curve in the representative force spectrum. However, we are thankful to the Reviewer for raising this concern. As of now, our automated force-curve sorting program (home-written using Matlab) did not have any filtering based on noise. We are in the process of including a filter based on MAD on the tail region during the free flight of the cantilever upwards with no string attached to the surface.

Next, we plotted the force-distributions with and without the noisy force curves and noticed no noticeable differences in the force-distributions (**Figure X(b-e)**). Thus, we appeal to the Reviewer and the Editor to allow to replace the representative force-curve with a new Figure 1.

Figure X: (a) Mean absolute deviation determined for all the force curves obtained in HAP of Cdh23 EC1-27 at $3000 \text{ nm}\cdot\text{s}^{-1}$ pulling speed of the cantilever. In the inset, representative curve possessing higher noise is shown. Arrowhead points towards the level of mean absolute deviation for the specific curve. Force distributions obtained for HAP of Cdh23 at pulling velocity of $3000 \text{ nm}\cdot\text{s}^{-1}$ by considering all the unfolding force-curves (black void histogram)

and removing the force curves having higher noise (red rectangle box in the histogram) for different length change of (b) 6.4 nm (c) 15.8 nm (d) 24.5 nm (e) 33.7 nm.

Figure 1 (Revised): Descriptions of HAP and PAP. (a) Schematic representation of HAP of Cdh23 using AFM where the C-terminus of Cdh23 is covalently attached to glass coverslip using sortagging. The N-terminal of Cdh23 is recombinantly modified with biotin and pulled by streptavidin-coated cantilevers. (b) Scheme of PAP configuration depicting the partner-assisted pulling of Cdh23 EC1-27 with Pcdh15 EC1-2 (red). (c & e) Representative force-extension curves of Cdh23 in HAP (black) at 2000 nm.s⁻¹ and 3000 nm.s⁻¹ pulling speeds, respectively. Both curves depict an initial long stretch followed by sawtooth patterns. (d & f) Representative force-extension features of Cdh23 in PAP (red) at 2000 nm.s⁻¹ and 3000 nm.s⁻¹ pulling speeds respectively. Both curves exhibit multiple unfolding peaks at the initial stretches, followed by sawtooth patterns of unfolding. The dotted green lines in c-f are the WLC model fit. (g & h) Distributions of ΔL_c of Cdh23-stretching in HAP (black line) (n=838 unfolding events) and PAP (red) (n=946 unfolding events), respectively. The data is combined for all loading rates. Solid lines represent the Gaussian fits exhibiting four unfolding peaks. (i) Distributions of initial long and short stretches of Cdh23 obtained in HAP (black) and PAP (red), respectively. The data are plotted for all the force curves obtained at different pulling velocities. Gaussian fittings of the histograms (solid black line) exhibit a mean length gain of 75.0 ± 2.2 nm in HAP and 6.5 ± 0.3 nm in PAP. Errors mentioned are the standard error of fitting. (j) Probability distributions of four different stretches of Cdh23 in HAP (black) and PAP (red).

2) It is still not clear why gaussian fitting of force distributions is done. The authors have mentioned that they have done it in a qualitative way to get the force peaks however it is not the case. The force values are used to extract other parameters of energy landscape for which correct distribution functions should be used. As the authors have mentioned the heterogeneity in the traces resulted in poor statistics then why energy landscape parameters were quantified. For single-molecule measurements statistical significance is must and thus the manuscript looks weak.

Reply: We agree with the Reviewer. Not all the force-distributions in our measurements follow perfect normal or lognormal distributions. Further, the force distributions span a high range of forces. We, therefore, performed a non-parametric Kernel Density Estimate (KDE) to obtain the continuous frequency distribution (Figure 2, a, b, c, d). The KDE has been optimized and widely used in the dynamic force spectroscopy in recent times (Mulhall et. al, Nat Commun 12, 849 (2021); Bura et. al, J. Chem. Phys. 130, 015102 (2009);). We, too, have determined the most-probable unbinding force (F_{mp}) from the kernel distribution's maxima. Subsequently, we used the Bell-Evans model to determine the energy landscape parameters. The kinetic parameters (off-rate, distance to the transition state, and free energy change) obtained using this method, though vary from the previous estimate, follow a similar trend

when compared between HAP and PAP (Table 1, attached below, and Table 1 in the main manuscript).

Figure 2. Unfolding nature of Cdh23 in HAP and PAP. The distributions of unfolding forces of Cdh23 at different pulling velocities (500, 1000, 2000, 3000, 5000 $\text{nm}\cdot\text{s}^{-1}$) have been plotted for four different extensions of (a) ~ 6.5 nm, (b) ~ 16 nm, (c) ~ 24.5 nm, and (d) ~ 33 nm for both HAP (black) and PAP (red). The solid and the dashed blue lines represent the kernel density estimates for HAP and PAP, respectively. The most probable unfolding forces are chosen from the maximum counts in the KDE. (e-h) The monotonous increase in the most probable unfolding forces with loading rates (loading-rate = pulling velocity \times spring constant of the cantilever) is plotted for four different ΔL_c s. Errors reported are the standard error of fitting. The solid lines are the Bell-Evans model fit to the force-loading rate data for (e) ~ 6.5 nm, (f) ~ 16 nm, (g) ~ 24.5 nm, and (h) ~ 33 nm. (i & j) Distributions of the intrinsic lifetime (τ_{off}) and x_β of all four extensions are shown respectively for HAP (black) and PAP (red) (mean \pm SD from $n=3$ independent experiments). (k & l) Schematics of potential energy barriers for extensions of Cdh23 of ~ 6.5 nm (black), ~ 16 nm (red), ~ 24.5 nm (green), and ~ 33 nm (blue) are drawn as per the kinetic parameters for HAP and PAP, respectively.

Table 1: (a) The kinetic parameters of all unfolding steps of Cdh23 EC1-27 in PAP mode. (b) The kinetic parameters of all unfolding steps of Cdh23 EC1-27 in HAP.

The mean±s.d. are estimated from n=3 independent experiments.

(a)

Extension (nm)	Lifetime (s)(mean±s.d.)	Transition state distance (x_β) (nm)(mean±s.d.)	Energy barrier (ΔG^*)	Fc (pN) = $\Delta G^*/x_\beta$
6.5±0.5	0.038±0.007	0.096±0.009	17.4 $k_B T$	181.2
16.9±1.4	0.027±0.003	0.054±0.005	17.1 $k_B T$	316.6
24.6±0.6	0.052±0.004	0.079±0.006	17.7 $k_B T$	224.0
33.2±0.9	0.061±0.004	0.074±0.005	17.9 $k_B T$	241.9

(b)

Extension (nm)	Lifetime(s) (mean±s.d.)	Transition state distance (x_β) (nm) (mean±s.d.)	Energy Barrier (ΔG^*)	Fc (pN) = $\Delta G^*/x_\beta$
6.4±0.4	0.026±0.004	0.086±0.008	17.1 $k_B T$	198.8
15.8±0.3	0.010±0.003	0.071±0.006	16.2 $k_B T$	228.2
24.5±0.3	0.039±0.003	0.108±0.007	17.5 $k_B T$	162.0
33.7±0.2	0.030±0.005	0.088±0.004	17.2 $k_B T$	195.4

3) If the binding strength of EC1-2 of Cdh23 and Pcdh15 is high, why the authors have not reported the forces required to rupture the interactions. These values should be mentioned in the text so that a direct comparison can be made with the unfolding forces of the domains.

Reply: We have measured the binding strength of Cdh23 EC1-2 vs Pcdh15 EC1-2 complex using AFM previously. Lifetime found using this study lies in the range of ~200 s. Lifetime for various unfolding types obtained in this study falls below 1 s. But, one of the essence of this work lies in the finding that unfolding of all non-interacting domains prevents breaking of the interaction. This leads to higher force withstanding capability by the two interacting domains. So, the force measured during the unbinding of the tip-link complexes formed by the two outermost domains only would not reflect the true scenario happening during the measurements of Cdh23 EC1-27 vs Pcdh15 EC1-2 interactions. However, lifetime obtained for the only interacting complex is many orders higher than the lifetime of each unfolding types obtained from our present study.

We have included the following statement in the last paragraph of the Introduction section:

To note, single-molecule force spectroscopy using AFM has obtained an off-rate of $4.5 \times 10^{-3} \pm 4.9 \times 10^{-5} \text{ s}^{-1}$ and distance to the transition state of $0.18 \pm 0.03 \text{ nm}$ of the tip-link complex when measured using the two outermost domains of tip-link cadherins.